# Artemether and Praziquantel: Origin, Mode of Action, Impact, and Suggested Application for Effective Control of Human Schistosomiasis

**DOI:** 10.3390/tropicalmed3040125

**Published:** 2018-12-19

**Authors:** Robert Bergquist, Hala Elmorshedy

**Affiliations:** 1Ingerod, SE-454 94 Brastad, Sweden; robert.bergquist@outlook.com; 2College of Medicine, Princess Nourah bint Abdulrahman University, Riyadh 11671, Saudi Arabia; 3Department of Tropical Health, High Institute of Public Health, Alexandria University, Alexandria 21561, Egypt

**Keywords:** schistosomiasis, elimination, praziquantel, artemether, combination therapy

## Abstract

The stumbling block for the continued, single-drug use of praziquantel (PZQ) against schistosomiasis is less justified by the risk of drug resistance than by the fact that this drug is inactive against juvenile parasites, which will mature and start egg production after chemotherapy. Artemisinin derivatives, currently used against malaria in the form of artemisinin-based combination therapy (ACT), provide an opportunity as these drugs are not only active against malaria plasmodia, but surprisingly also against juvenile schistosomes. An artemisinin/PZQ combination would be complementary, and potentially additive, as it would kill two schistosome life cycle stages and thus confer a transmission-blocking modality to current chemotherapy. We focus here on single versus combined regimens in endemic settings. Although the risk of artemisinin resistance, already emerging with respect to malaria therapy in Southeast Asia, prevents use in countries where ACT is needed for malaria care, an artemisinin-enforced praziquantel treatment (APT) should be acceptable in regions of North Africa (including Egypt), the Middle East, China, and Brazil that are not endemic for malaria. Thanks to recent progress with respect to high-resolution diagnostics, based on circulating schistosome antigens in humans and molecular approaches for snail surveys, it should be possible to keep areas scheduled for schistosomiasis elimination under surveillance, bringing rapid response to bear on problems arising. The next steps would be to investigate where and for how long APT should be applied to make a lasting impact. A large-scale field trial in an area with modest transmission should tell how apt this approach is.

## 1. Background

The World Health Organization (WHO) includes schistosomiasis among the neglected tropical diseases (NTDs) and has selected it for elimination [1]; this may be premature as there is still a large discrepancy between the number of people requiring preventive chemotherapy (PCT) and those presently receiving it [2]. The disease is caused by trematode parasites with a predilection for abdominal capillary veins in the mammalian definitive host. Various freshwater snails act as intermediate hosts for the six different species capable of infecting humans. Out of these, *Schistosoma mansoni*, *S. haematobium*, and *S. japonicum* together cause almost all cases of schistosomiasis, which amounts to about 240 million infected and 700 million at risk worldwide [3]. The prevalence is often focal with high transmission in some spots and none in others and the lesions caused, intestinal or urogenital depending on the species of the parasite, tend to be chronic rather than acute. The attributes of infection include poor sanitation, proximity to water bodies, type and extent of water contact, snail population density, as well as age and gender of the human population in the endemic areas. For example, the prevalence and intensity of infection are significantly higher in males and in school-age children [2,4,5]. Theoretically, successful schistosomiasis control should be possible through a multidisciplinary approach focusing on chemotherapy, snail control, provision of water, sanitation, and hygiene (WASH) as well as behavioural change [6,7]. Nevertheless, elimination will be difficult to achieve in practice, and very few endemic countries have been entirely successful in controlling the disease. Interestingly, Japan managed to eradicate the infection (in 1977), mainly thanks to snail control based on environmental management, as described by Tanaka and Tsuji [2,4,5]. 

The cornerstone strategy for schistosomiasis control was for a long time directed at its snail intermediate host using broad-spectrum molluscicides. However, when praziquantel (PZQ) was introduced [8] and soon afterwards used for mass drug administration (MDA), it not only replaced other drugs thanks to safety, high efficacy and easy administration [9,10,11] but effectively became the only approach. This modality also changed the focus from snail control and infection prevention to morbidity reduction, reflected in a declining disability-adjusted life years (DALY) metric for schistosomiasis [12,13].

In China, Chairman Mao Zedong, not only instigated the National Schistosomiasis Control Programme [14] but also took an interest in malaria. The launch of the malaria research programme (under code name 523) in 1967 focused on herbs used in traditional Chinese medicine. Working in this programme, Youyou Tu, soon identified the ingredient artemisinin (qinghaosu) in extracts from wormwood (*Artemisia annua*) as a powerful antimalarial [15,16]. Qinghaosu has been used in China for more than 2,000 years for the treatment of fevers, and the story of how it was discovered has eloquently been told by Faurant [17] and Miller [18]. Already in 1971, Chinese chemists isolated the active lactone with its peroxide grouping [19]. However, this knowledge did not reach the West until the late 1970s when it was mentioned in a Welcome News Supplement [20]. Youyou Tu’s own remarks [15] at the Fourth Meeting of the WHO Scientific Working Group on the Chemotherapy of Malaria, held in Beijing 1981, provided further details of the new drug. The landmark work carried out under her direction [21] constituted a break-through for malaria therapy and led to the development of artemisinin-based combination therapy (ACT) that has since revolutionized the care of malaria patients [22] eventually leading to Youyou Tu being awarded the Nobel prize in Medicine for 2015, together with Campbell and Ōmura for ivermectin [23,24]. However, the story does not end there since, amazingly, it was found that the artemisinin do not only affect the malaria parasites, but are also active against juvenile schistosomes, which was first shown by Chen et al. [25] at the end of the golden decade of antiparasitic drug discovery in the 1970s. In fact, this discovery predates that of scholarly articles on qinghaosu’s use against malaria, which remains its main application. 

## 2. Pharmacological Aspects

Single-celled malaria plasmodia and schistosome worms are phylogenetically very distant from each other, and one might think that the damage caused by the artemisinins to both these organisms would be due to different principles. However, there are reasons to believe otherwise, considering that both parasites feed by the digestion of haemoglobin in host erythrocytes, leading to a surplus of haeme (protoporphyrin-IX with a ferric ion at its centre) or haemin (ferriprotoporphyrin-IX characterized by an extra chloride ion bound to the ferric ion), both of which produce destructive hydroxyl radicals capable of provoking the alkylation of parasite proteins [26]. Faced with this threat, both parasites detoxify these compounds by crystallizing them into the inert polymer haemozoin (ferriprotoporphyrin-IX) that is found in quantity, both in *Plasmodium* [27] and adult schistosomes [28]. It seems that haeme and haemin are implicated in the destruction of *Plasmodium* and *Schistosoma* due to interference with the formation of hemozoin. Various drugs besides artemisinins (e.g., trioxalanes [29,30,31], trioxaquines [32], and ozonides [33,34]) are also capable to form alkylates with free haeme and haemin and use them for further alkylation, this time between parasite proteins. Indeed, this seems to be an example of independent parallel evolution making chemical pathways available when needed, thus appearing de novo without connection to previous genetic information. 

### 2.1. The Artemisinins

The active principle in qinghaosu is a lactol endoperoxide, which was used to produce artemether (ART) and artesunate, two semisynthetic artemisinin derivatives which become active after being metabolised in the blood into dihydro-artemisinin [35]. The current working hypothesis is that this drug acts through haeme-dependent reduction to sequentially generate free radicals: the haeme iron first attacks and breaks the endoperoxide linkage to artemisinin, producing an oxygen-free radical, which is then rearranged to produce a carbon-free radical that causes lethal damage through the alkylation of parasite proteins [36]. Work by Xiao et al. [31,37,38,39] and Chaturvedi et al. [36], further developed and summarized by Xiao and Sun [26], indicates that the same pathway is followed with respect to *Schistosoma*. Antioxidant enzymes systems available in adult worms, but not common in immature ones, can prevent this effect. However, to be effective, the drug needs to be ingested by the parasite, enabling the interaction with haemin or haeme causing damage to the worm gut by generating one or many substances toxic to these worms in amounts overwhelming the pathway leading to the hemozoin. The fact that the gut suffers particularly severe damage after ART treatment supports this chain of events [37]. A disadvantage of the artemisinin derivatives is their short in vivo half-life, typically ~2 h in humans [40].

### 2.2. Praziquantel 

In spite of being used for 40 years, and most of that time in the form of MDA, the exact molecular mechanism of PZQ remains unknown. However, the drug has no effect on the enzymes discussed above but relies on a rapid influx of Ca^2+^ into the worm (interestingly, immature forms are refractory), leading to intense muscular paralysis together with damage to the tegument [41,42]. How this disruption is linked to the original binding of the drug is still unknown, but it is thought that the exposure of parasite surface antigens leads to recognition and parasite clearance through immunological means, something which may indirectly account for the difference in sensitivity between juvenile and mature stages [43]. Although the receptor is not known, it has been shown that PZQ disrupts ion transport and recent experimental evidence indicates that transient potential (TRP) channels are targeted by the drug; this could result in the increased permeability of adult schistosome cell membranes towards calcium ions [44]. 

### 2.3. Combination Treatment

With artemisinins active against immature schistosome worms [25,45,46] and PZQ primarily targeting adults [47,48], the combination of PZQ and ART presents a chemotherapeutic perfect storm against this parasite as shown in Figure 1, where the y-axle shows the cidal effect expressed as percentage of killed parasites in the experimental animals used. Although small susceptibility variations with respect to parasite age can be seen between the species, there is also a short period coinciding with a larval age between 4 and 5 weeks when all three species are partially refractory against both ART and PZQ. However, the main difference between the two drugs is that the former is predominantly active against juvenile stages and the latter against adult stages. In addition, although PZQ shows a similar activity against *S. haematobium* and *S. japonicum* worms, there is quite a different picture when given to *S. mansoni,* as it obviously also has some activity against juvenile stages. 

*S. mansoni* reactions to ART according to experiments by Xiao et al. [46] and to PZQ by Gönnert and Andrews [48], were confirmed by Sabah et al. [47]; *S. japonicum* to ART by Xiao et al. [39] and to PZQ by Xiao et al. [49], were confirmed by Wu et al. [50]; and *S. haematobium* to PZQ by Botros et al. [51]. Note that *S. haematobium* reactions to ART represent indirect information on parasite susceptibility based on tegumental damage [52,53] and cannot therefore be directly compared with other measurements presented in this figure.

## 3. Drug Resistance

### 3.1. Artemether

Resistance to artemisinins in malaria parasites, defined as a slower rate of parasite clearance in patients under treatment, has emerged in Southeast Asia’s Mekong region [54]. The parasites are thought to mount a defensive stress response, and recent evidence suggests that this response in certain mutants is enhanced, thus promoting parasite survival leading to drug resistance [55]. These authors have shown that the enhanced cellular defence response that underlies resistance development enables very early ring stages to withstand drug exposure for longer. However, in this case, the intrinsic sensitivity to artemisinin is retained [55], which is an encouraging sign that might also play a role in schistosomiasis therapy in due course, although there is as of yet no knowledge about this. On the other hand, drug resistance may still appear in the longer perspective, and might well do so if the drug were to be incorporated in widespread control schemes. It should also be said that the determination of whether or not resistance has developed is not straightforward when the drugs are used against schistosomiasis as the parasites are localized in abdominal veins, that is, in areas where they cannot be directly observed.

A major problem preventing the use of ART for schistosomiasis control is the increased risk for the spread of drug resistance against malaria. This is very clearly a big risk, and the use of the drug against schistosomiasis should be restricted to areas outside those where there is any trace of malaria transmission. As can be seen in Figure 2, there are such areas—in China, Brazil, Mediterranean Africa including Egypt, and the Middle East—where a combination treatment would be particularly useful at the elimination stage.

### 3.2. Praziquantel

Drug resistance, to be expected after long-term use of repeated, extensive application, is a clear risk, but evidence with respect to PZQ remains scant. However, if it were to emerge, current efforts to eliminate schistosomiasis would be severely challenged requiring alternative and/or complementary drugs [56,57]. Ominously, drug resistance against PZQ has been generated in the laboratory following treatments of *S. mansoni* strains with subcurative drug doses [58,59,60,61], proving that it cannot be ruled out. Furthermore, resistant *S. mansoni* isolates have been reported from various African sites (Egypt and Kenya in particular [62,63,64]), while several cases of failed parasite clearance following a standard treatment for *S. haematobium* infections in Africa have also been shown [65,66,67]. In China, the emergence of drug resistance has also been experimentally produced [68,69], including reports of resistant *S. japonicum* isolates [70]. Based on the evidence referred to above, it can be concluded that drug resistance in the field is a clear possibility and it is just a question of how fast it will spread once established.

Even if the influx of calcium ions into the worm after exposure to PZQ is regularly observed, the molecular mechanism of PZQ’s mode of action is not fully unravelled. The lack of widespread resistance after long-term use may be due to the presence of multiple pharmacologically relevant targets, and Thomas and Timson [71] suggest that PZQ may act, at least in part, on a protein–protein interaction and that altered drug metabolism or enhanced drug efflux are the most likely ways resistance may arise. Additionally, laboratory tests to identify resistance to PZQ have demonstrated a specific region in the parasite genome which might be responsible for reduced drug susceptibility [72].

## 4. Trials and Community-based Studies 

Although annual or biennial MDA with PZQ controls schistosomiasis morbidity well, even in high-transmission areas, it does not generally achieve any significant reduction in transmission [73]. ART, on the other hand, has been shown to do so in several randomized control studies [74,75,76,77,78], at least during the limited time covered by the studies. With regard to single-drug trials using artemether, Utzinger et al. [79] investigated the use of ART for prevention of *S. mansoni* infection using a randomized, double-blind, placebo-controlled trial in Côte d’Ivoire, demonstrating that ART resulted in a significantly lower incidence of infection (24% versus 48.6%, relative risk: 0.50 [95% CI 0.35–0.71], *p* = 0.00006), translating into a 50% risk reduction. A follow-up study on the application of ART against *S. haematobium* in the same country found similar results, although the protective efficacy was considerably lower [78]. A few studies more similar to interventions have been carried out. In Nigeria, for example, PZQ combined with artesunate was used for the treatment of urinary schistosomiasis in 312 randomly selected schoolchildren, reaching a cure rate of 89%, something statistically significant. Two groups, given either monotherapy with PZQ and artesunate to compare, showed a 73% and 71% cure, respectively, demonstrating that the use of combination treatment is both safe and more effective than treatment with either drug alone [76].

Results of a systematic reviews and meta-analysis of a large number of clinical trials have demonstrated the superior performance of the combined drug regimens versus single-drug application, as shown by Wu et al. [45], Liu et al. [35] and Perez del Villar et al. [80]. The first two reviews concerned *S. japonicum* infections only, out of which the former reports ten randomized controlled trials with participants ranging from 318 to 5098; four were multi-centre studies, and six were carried out at single centres. Artemisinin compounds (artesunate or ART) had few side effects and were found to be effective at 7-day and 15-day intervals (preferentially when used at 15-day intervals) for preventing infection during short-term exposures, such as during flooding relief work [45]. Liu et al. [35] state that protection was considerably higher when using combination therapy (84–97%), while it was only 52% with monotherapy (PZQ at 40 mg/Kg), while increasing the dosage to 60 or 100 mg/Kg resulted in a protection rate up to 91%. Interestingly, the protection rate of artemisinin derivates alone exceeded that of PZQ alone with protection rates reaching 97%, the highest rates achieved when the number of doses (3–8) were increased and the interval between them shortened from 1 month to 1 week. Confirming these results, not only for *S. japonicum* but also for *S. haematobium* and *S. mansoni*, Perez del Villar et al. [80] also reported better results with combination therapy than that achieved using either drug alone. In addition, the reviews convey the impression that ART, rather than artesunate, is the preferable choice when combined with PZQ. 

In Egypt, good PZQ coverage has resulted in a substantial drop of infection intensities in most endemic settings [81,82,83]. However, despite a regular distribution of PZQ since almost 20 years, transmission continues at an appreciable and unacceptable level in many foci, especially in the Nile Delta. The experience in Kafer El-Sheikh Governorate, located close to the end of the Rosetta branch of the Nile River in the northern part of the Delta, is one of the hotspot areas with regard to transmission that clearly reflects the failure of controlling schistosomiasis with PZQ alone. As shown by Barakat et al. [81,82] and Haggag et al. [84], the prevalence and intensity of infection in children newly enrolled in primary schools remain almost unaffected [75]. In addition, premature interruption of MDA programmes due to the insensitivity of routinely used diagnostic tests often result in the re-emergence of infection in the span of a few years with concern that the prevalence and intensity of infection might again bounce back to higher levels.

Our own field experience in Kafer El-Sheikh Governorate have clearly demonstrated the remarkably positive effect of giving ART and PZQ in combination [74]. We conducted a double-blind, randomized controlled trial with 913 children, in which two groups (experimental and controls) received 40 mg/kg body weight of PZQ twice, four weeks apart at baseline. Afterwards, the experimental group received 6 mg/kg body weight of ART every 3 weeks in 5 cycles during the transmission season, and the control group received a placebo. At the end of the study, prevalence of infection in the group receiving ART was approximately half that of the placebo group: 6.7% versus 11.6%, and incidence of new infection for the ART group was 2.7% versus 6.5% for the placebo group, i.e., a clearly significant risk reduction. What was unique about this study is that we treated the patent infection with two doses of PZQ 4 weeks apart, 40 mg/Kg each to maximize the cure rate, thereby achieving the aim to evaluate the prophylactic effect of ART alone. It should be added that five annual rounds of PZQ treatment with a coverage rate above 90% in the same area was shown to have reduced the incidence from 29% to 12 %, while the incidence dropped from approximately 19% in 2000 to 12% in 2001 [85]. Another study including 2382 individuals of both sexes and all ages in five high-risk Nile Delta villages in Kafer El-Sheikh showed an overall prevalence of schistosomiasis of 29% with a generally low intensity of infection. Although this result was deemed positive compared to the situation before MDA using PZQ was implemented, the outcome was tempered by the insight that transmission remained largely uninterrupted after long-term uninterrupted control activities [75]. To ensure better cure rates in this area, prevent rapid re-infection, and avoid the potential development of drug resistance, the use of ART/PZQ combinations should be instituted. 

## 5. Discussion

Treatment failure can be due to drug resistance against PZQ, but it can also be due to a lack of therapeutic efficacy of the drug with respect to juvenile trematode flukes as shown by Doenhoff et al. [86]. In this connection, it should also be considered that PZQ, though it has a strong curative effect, never reaches 100% efficacy. These facts together reveal that this drug is not transmission-blocking, making the goal of elimination of schistosomiasis illusory if not combined with other drugs or tools, such as snail control, health education, water, sanitation, and hygiene (WASH). Overall, complementing PZQ with ART would not only target all stages of the parasite, from its penetration into the host (when PZQ has an ultra-short activity to the organism covering only a few hours) over its first weeks of growth to the adult stage. Importantly, by targeting the stages before egg production has started, the artemisinin derivates are truly a transmission-blocking drugs. In addition, its already high efficacy can be raised to almost 100% by using several doses provided on a weekly basis [46]; however, this approach would not be realistic in practise. This is of course only acceptable for short periods, when prevention is needed for which it has successfully been applied in flooding interventions in China [35,45]. Prolonged treatment periods have not been tried. 

The successful outcome of 40 years of schistosomiasis chemotherapy using PZQ, most of this time in the form of MDA is of a magnitude relieving morbidity and suffering of millions of people. This accomplishment has led to thoughts of worldwide elimination of the disease in the next decade, though others feel that it might take longer [87]. The stumbling block is that PZQ does not block transmission. This can, however, be achieved by conferring a transmission-blocking modality to current chemotherapy through the addition of artemisinin derivatives that act before the infection results in adult worms capable of producing eggs. However, due to the risk of drug resistance, which is already emerging in Southeast Asia, APT cannot be recommended in areas where ACT is needed for malaria care. On the other hand, China, Mediterranean Africa, and also large areas elsewhere where schistosomiasis and malaria do not overlap are also locations where the elimination of schistosomiasis would have the best chance of rapid success. As side effects of an ART treatment are mild or non-existent, we propose a precision treatment approach, involving repeated ART treatments (monthly or bimonthly) backed up with PZQ biannually until elimination has been achieved. However, the elimination of schistosomiasis from sub-Saharan Africa cannot depend on artemisinin derivates, due to the need to reserve these drugs for malaria treatment; it will therefore require more complicated, long-term schemes, preferably including a vaccine. It is encouraging that, thanks to high-resolution diagnostics, both for humans [88] and snails [89], it should be possible to keep areas scheduled for elimination under surveillance so that rapid response can be raised whenever needed [90]. 

Thanks to donors supporting purchase and distribution of PZQ (e.g., the Schistosomiasis Control Initiative working in Africa (http://www.imperial.ac.uk/schistosomiasis-control-initiative), the drug been made available free of charge in all large endemic areas. However, as pointed above, there is still a shortfall with respect to PZQ [2]. The addition of ART would raise the cost further but before requesting the extra funds, more convincing data is required. Although the field trials mentioned here support combination therapy, testing in conjunction with more sensitive and quantitative diagnostic tools will be needed to take us closer to the goal of applying ART/PZQ therapy for schistosomiasis elimination in non-malarious areas. Indeed, such diagnostic assays [91,92,93] have already shown that the extent of schistosome infection has been greatly underestimated, due to the diagnostic deficit of stool examination and urine filtration which are still commonly used in the endemic areas [94]. 

## 6. Conclusions

The obvious problem with PZQ is that it does not block transmission. Snail control has been tried with questionable results since the snails hosts of *S. mansoni* and *S. haematobium* are non-amphibious and can survive long dry periods. The outcome in China has been better, although not completely successful there either, since the intermediate snail host of *S. japonicum* is amphibious and therefore slightly easier to control. Other underlying factors favouring transmission are ecological conditions, poor sanitation, and the high intensity of unprotected water contact. Under such circumstances, additional control measures need to be adopted. One of these measures would be to change the current MDA strategy to include a combination of PZQ and ART in certain areas based on the rationale that it would confer a transmission-blocking modality to current chemotherapy. thereby ensuring higher cure rates, reducing (possibly even preventing) transmission and rapid re-infection, and avoiding a potential development of resistance to either drug. 

The spatiotemporal risk for reinfection dominates in endemic areas, which in principle would require keeping people on constant chemotherapy, at least for a period of time. Although this is not a realistic approach, a strategy consisting of PZQ provided at 6-month intervals together with a monthly ART treatment might be the recipe for not only achieving elimination, but in fact also making local eradication possible. It remains to be investigated how long APT regimens should last and how realistic this approach would be.

## Figures and Tables

**Figure 1 tropicalmed-03-00125-f001:**
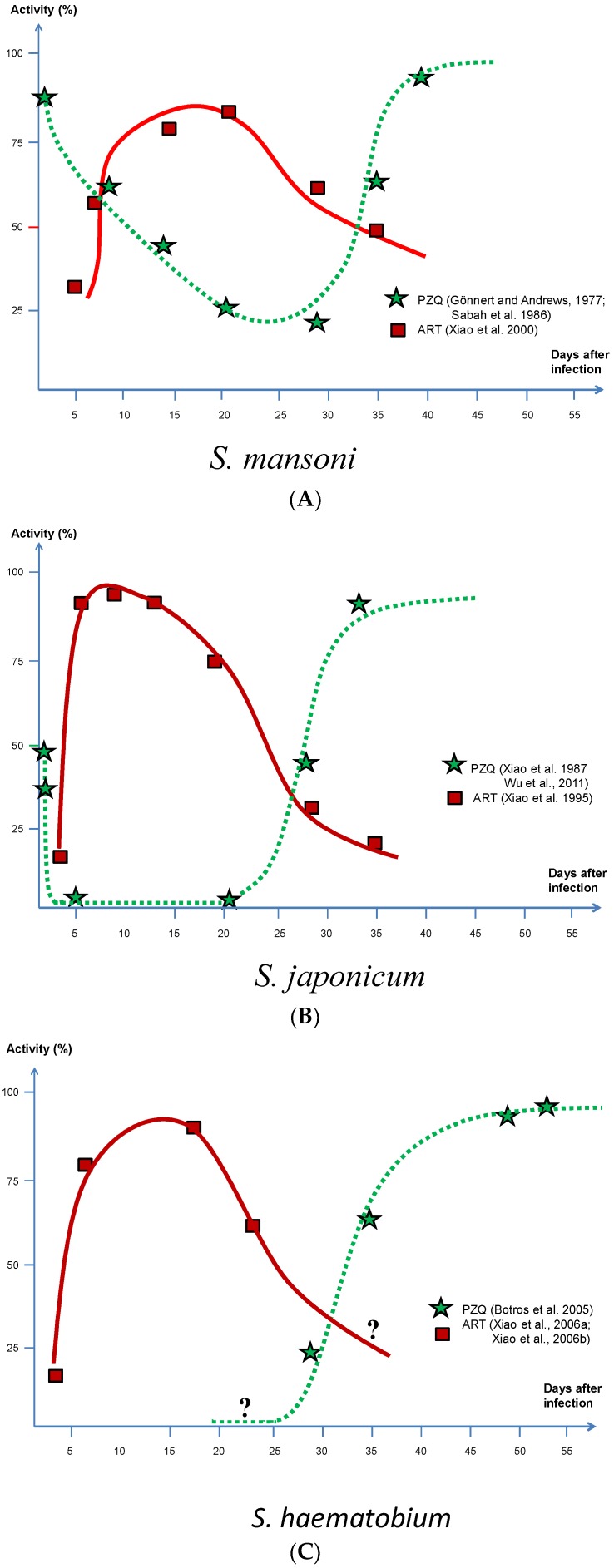
Activity of artemether and praziquantel in relation to different stages of maturity in all three main schistosome species. (**A**) *S. mansoni*; (**B**) *S. japonicum*; (**C**) *S. haematobium*.

**Figure 2 tropicalmed-03-00125-f002:**
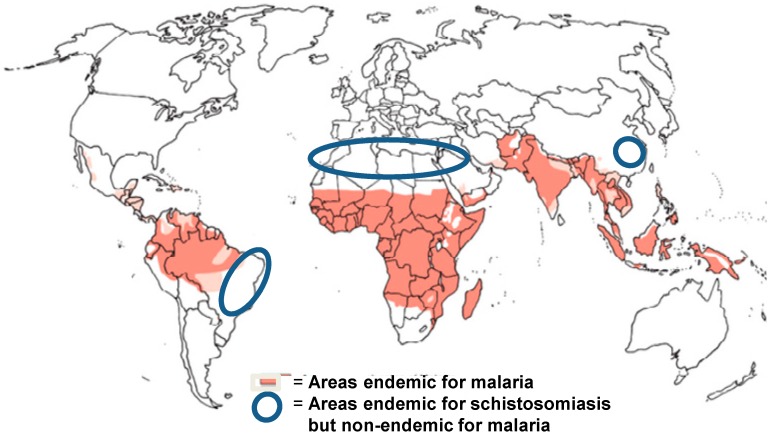
Areas where artemisinin derivates can be used for transmission interruption of schistosomiasis.

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
