# Peer review of "Artemether and Praziquantel: Origin, Mode of Action, Impact, and Suggested Application for Effective Control of Human Schistosomiasis"

_tropicalmed, 2018, doi:10.3390/tropicalmed3040125_

Round 1

Reviewer 1 Report

Line 15: 'provide an opportunity as these drugs'

Line 17: versus combined drug regimens in endemic settings'

Line 24: remove currently contemplated

What about costs of providing cmbo therapy? At the moment most PZQ is provided free for schisto control, what would be the deal with Artemether included?

Line 31: Not a huge fan of this opening sentence. 'Even if claimed' - well it is classified as an NTD so it's not about 'if' it's claimed. I kind of get what you mean, that it's a neglected disease but it's a huge deal in the endemic areas - although I would argue that knowlege of schistosomiasis in endemic areas is very poor among lay persons and even health professionals.

Line 32: 'endemic areas. There is a large'

Line 37: Italics on and between species names

Line 37: 'almost all cases of schistosomiasis, which'

Line 46: Although Japan managed it pre chemotherapy

Line 51: Define Praziquantel as PZQ here at first mention

Line 65: space between 1971 and isolated

Line 64-67: This is a confusing sentence, please consider rewriting. Perhaps just switch the sentence around, 'Chinese chemists isolated blah blah blah in 1971. However this knowledge did not reach the West until the late 1970s'

Line 67: This sentence doesn't really add anything, it's a meh sentence. Remove or make useful. For instance, provides evidence on....presumably the drug

Line 93-95: Well, rather than Nature or an act of God, perhaps the scientific explanation of parallel evolution

Line 112: 'The fact that the worm gut suffers'

Line 118: PZQ used here without previously defining it. First mention in line 51, define there then use abbreviation throughout

Figure 1: S. mansoni PZQ data is from reasonably old sources, is there nothing more recent?

Line 152: Could define South East Asia as (SEA)

Line 156, Line 158, Line 159: However, however, however. Three sentences in a row starting with however. 

Line 159: 'respect to Schistosoma.'

Line 161: 'resistance has developed'

Line 165: remove comma after problem

Figure 2: The Philippines not marked for schisto

From this section onwards (3.2 Praziquantel) there is a lot of going between using PZQ and praziquantel. Should be consistent. Either use the abbreviation or don't use it at all. Ditto with artemether which was defined as ART line 99

Section 4: So if the cure rate for PZQ is so low with PZQ is that underdosing or resistance? Under dosing would help lead to resistance, so should control programs just be dosing better? Are diagnostics used in control programs enough to determine parasite clearance? Does anyone even look to see if cure has been affected in MDA programs when lab scientists are not involved? 

Line 209: 'with PZQ or ART'

Line 222-223: So, underdosing? 40mg is recommended but efficacy is increased with increased dose. Back to previous point, should control and MDA programs just be dosing better.

Line 226, 233, 252, 257: double spaces occurring after full stop

Line 226-227: Species names not in italics 

Line 229: 'when combing with PZQ'

Line 240: Would this not still occur in combination therapy? Unrelated to drug choice

Line 251: Was this significant?

Line 253: space between 40 and mg

Line 257: space between 2001 and [unpublished

Line 267: Now we have PZQ and praziquantel used in the same sentence, with PZQ first

Line 277: Is multiple dosing going to achievable in endemic areas?

Line 283 and 290: Both MDA and APT have been previously defined in the introduction

Line 304: APT?

Line 313: 'schistosomiasis caused by infection with S. manosoni and S. haematobium' - also, how and why is nail control less effective? Or at least add reference which will state why.

Conclusion seems a little repetitive from last paragraph of discussion, consider combing that para into conclusions.

References:

Ref 1 has (accessed on      then nothing

ref 4 italics on species name

ref 7 italics on species name

Seem to change between capitalising every word and not doing that. example ref 13 and ref 14. 

ref 26 italics on genus name

ref 32: italics on species name

ref 34: italics on species name

ref 37: italics on species name

ref 42: italics on species name

ref 46: italics on species name

ref 47: italics on species name

ref 49: italics on species name

ref 51: italics on species name

ref 52: italics on species name

ref 53: italics on species name

ref 54 has (accessed on      then nothing

ref 55: italics on species name

ref 58: italics on species name

and so on and so forth

Author Response

 Thank for your  interest in our paper and the useful comments made.  We have responded to all queries and advice and incorporated them into the manuscript. We apologize for not being more strict regarding the use of italics and abbreviations. This has all been done.  We have also made other changes  marked them with yellow.

In addition, we have some further comments:

Regarding Figure 1., we could not find more up-to-date data for S. mansoni and PZQ

 We prefer spelling out Southeast Asia as it as it was not used many times, besides SEA is a commonly used abbreviation for Schistosoma egg antigen.

 Thank you for pointing out the problem with Figure 2, which caused a misunderstanding. We wished only to mark areas endemic for schistosomiasis and non-endemic for malaria. The legend in the figure has now been changed accordingly

Unfortunately, there are no checks on cure after MDA, if this would be done with the new POC-CCA test or even better with the CAA test, we would know more about dosage. Hopefully this will be done; however, this is beyond what we aimed for in this paper.

Thank you

Reviewer 2 Report

Please refer to attached files.

Comments:

Major:

1.    Line 240: Is the premature interruption of MDA due to insensitivity of routinely used diagnosis as stated or is it due to other factors such as logistics problems, acceptance of target population etc.? Please clarify. 

2.    Lines 278-280: What could be the implications of prolonged combination treatment?

3.    Lines 293-295: How long will the precision treatment approach be? What are the possible side effects of this combination therapy and acceptability of this in endemic areas?

Minor: (Typo errors) highlighted in the attached pdf

1.    Line 65: space between 1971 and isolated

2.    Line 84: delete d from evidenced

3.    Line 223: space between or and 100

4.    Line 313: insert to between due and S. mansoni.

5.    Line 470: capitalize Schistosoma and italicize Schistosoma haematobium

6.    Italicize the highlighted words in lines 104, 201, 215, 221, 226, 238, 269, 337, 346, 399, 416, 423, 444-445, 455, 457, 461, 467, 473-474, 480, 488, 491-492, 494, 497, 500, 506, 509, 512, 515, 523, 524, 529, 533, 534, 563, 566, 580, 590,  and 594.

7.    Shouldn’t the first letter of the journals be capitalized?

Author Response

Thank you for your  interest in our paper and the useful comments made.  We apologize for not being more strict regarding the use of italics and abbreviations. We have responded to all queries and advice and incorporated them into the manuscript. Changes are highlighted in blue in the revised manuscript. 

Thank you 

Reviewer 3 Report

The main thrust of the review by Bergquist and Elmorshedy is that combination therapy with artemether and praziquantel will give a better outcome in the control and elimination of schistosomiasis than the use of praziquantel alone. They rightly qualify this by saying that such a therapy should only be applied in regions where malaria is not endemic since artemether is one of the mainstays of malaria control worldwide. Its over-use in malaria-endemic regions could hasten the development of drug resistance. Whilst I agree with these sentiments, I feel the tone of the text is rather optimistic about the possibilities for disease eradication, and might strike a more realistic note in this respect.

Specific points for consideration:

Abstract: Use of the term “transmission blocking”. I struggled with this term. I guess what the authors are implying is that artemether prevents adult worm establishment after infection whereas praziquantel only kills adult worms. That’s a very fine point as the worms do not multiply in the host so whether you eliminate them as juveniles or adults, they are still removed by the treatment. I think a better selling point is that the combination therapy kills two different life cycle stages so would be complimentary, and potentially additive.

P3, Mode of action of artemisinins. One advantage claimed for artemether is that it will kill juvenile schistosomes whereas praziquantel is only active against adult worms. It is believed to act by preventing detoxification of haem, produced by digestion of haemoglobin, into haemozoin crystals. On line 107 the authors state “the only difference being that Plasmodium is only at risk during the blood stage of the malaria infection, while Schistosoma is constantly immersed in blood”. Larval schistosomes spend about four days in the skin before entering the bloodstream, while feeding on red blood cells only begins around ten days when the first worms reach the intrahepatic vessels of the portal vein. So just being in the bloodstream should not of itself endanger the worms – they must swallow red cells to become vulnerable. I believe that artemether has a very short life in the bloodstream so the inception of blood feeding around day 10 might to some degree explain why the drug has very little activity when administered 2-5 days post infection (See Fig 1).

P6, line 196. Trials and community-based studies. The authors note that “Although …… praziquantel controls schistosomiasis morbidity well even in high-transmission areas, it does not generally achieve any significant reduction in transmission”. I am happy to accept that inclusion of artemether improves on the situation by further reducing prevalence and incidence. However, this brings me to a significant weakness of the review. The cited study in Egypt (Ref 74) used only fecal smears to assess the intensity of infection. It has become inescapable that the extent of schistosome infection has been greatly underestimated due to the diagnostic deficit. The development of more sensitive diagnostics is alluded to in the Abstract (line 21) but not given serious consideration elsewhere. If the projected trial of artemisinin-enforced praziquantel treatment is to have real value it needs to be conducted in the context of the best diagnostic tools available, and certainly not rely on fecal smears. A list of relevant recent references on diagnostics is given below – see especially Colley et al., 2017.

Used in conjunction with sensitive and quantitative diagnostic tools, the combination artemether/praziquantel therapy might well take us closer to the goal of schistosomiasis elimination in non-malarious areas.

Corstjens PL, et al., (2014) Tools for diagnosis, monitoring and screening of Schistosoma infections utilizing lateral-flow based assays and upconverting phosphor labels. Parasitology 141(14):1841-55.

Weerakoon KGAD, et al., (2015) Advances in the diagnosis of human schistosomiasis. Clinical Microbiology Reviews 28 (4), 939–967

Kittur N, et al., (2016) Comparison of Schistosoma mansoni prevalence and intensity of infection, as determined by the circulating cathodic antigen urine assay or by the

Kato-Katz fecal assay: a systematic review. The American Journal of Tropical Medicine and Hygiene 94 (3), 605–610.

Colley DG, et al., (2017) Schistosomiasis is more prevalent than previously thought: what does it mean for public health goals, policies, strategies, guidelines and intervention programs? Infectious Diseases of Poverty 6, 63.

Ortu G, et al., (2017) Countrywide reassessment of Schistosoma mansoni infection in Burundi using a urine-circulating cathodic antigen rapid test: informing the national control program. The American Journal Tropical Medicine Hygiene 96, 664–673.

Ogongo P, Kariuki TM, Wilson RA. (2018) Diagnosis of schistosomiasis mansoni: an evaluation of existing methods and research towards single worm pair detection.

Parasitology 145(11):1355-1366

van Grootveld R, et al., (2018) Improved diagnosis of active Schistosoma infection in travellers and migrants using the ultra-sensitive in-house lateral flow test for detection of circulating anodic antigen (CAA) in serum. Eur J Clin Microbiol Infect Dis. 37(9):1709-1716.

Author Response

Thank you  for your  interest in our paper and the useful comments made.  We have responded to all queries and advice and incorporated them into the manuscript. Changes are highlighted  with  green.

Best regards

Reviewer 4 Report

The subject discussed in this review contains matters related to development of the new control strategies for human schistosomiasis.  Since the discovery of strong efficacy of artemisinin- derivatives against both malaria and schistosomiasis, it has been discussed about the probability of application of artemisinin-based combined therapy for the two important tropical diseases.  This review is concise but enough comprehensive for considering schistosomiasis control activity in the new era.  It is a review article, however, additional information should be covered and detailed  description for some critical parts are needed.

Major points

(1) As background information, ACT (but not APT) has already been tested for efficacy against human schistosomiasis in several areas endemic for both malaria and schistosomiasis (such as Obonyo CO et al. Lancet Infect Dis 10:603, 2010) .  ACT had no effects for human schistosomiasis in several reports published.  If mephloquine was the counter drug of artemisinin, it was not easy to interpret the results because both artemisinin and mephroquine have anti-schistosome effects...  It seems to be more informative for the Journal readers to understand such the historical background for the present topic.

(2) To make the point more clear about enhancing the risk of artemisinin-resistant malaria parasites, it is better to give the tentative optical therapeutic doses of artemisinin-derivatives for malaria and schistosomiasis.

(3) The concept emphasized in this manuscript is the difference of developmental stages susceptible to praziquantel and artemisinin-derivatives.  It is not easy to understand why artemisinin-derivatives have prophylactic effects through affecting the juvenile stage parasites.  

Minor suggestions

(1) What does the vertical bar of Fig 1 mean?  What is "Activity (%)"?

(2) The paragraph, Line 197 - 211, is complicated.  It is better to summarize in a Table.

(3) Line 209 - 210: Were there significant differences among "89%", "73%", and "71%"?

(4) In addition to Ref. #88, papers by Kumagai et al, Am J Trop Med Hyjg 83:542, 2010, or Tong et al, Acta Tropica 141:170, 2015 should be referred.

(5) Lines 226 - 227: italic description for the Genus and Species should be used.

Author Response

Thank you for your  interest in our paper and the useful comments made.  We have responded to all queries and advice and incorporated them into the manuscript. We apologize for not being more strict regarding the use of italics and abbreviations. This has all been done.  We have also made other changes and highlighted them with  grey.

 In addition, we have some further comments:

1. Regarding the suggestion to take into account the paper by Obonyo et al. Lancet Infect Dis 10:603, 2010, we felt that it would be beside out aim as we focus on combination treatment with PZQ and ART, while this paper investigates artesunate with sulfalene plus pyrimethamine versus praziquantel for treatment of Schistosoma mansoni.

2. Regarding ref. #88, it was suggested to include refs. to Kumagai et al, and Tong et al. We agree and these authors are indeed referred to in paper #88..

3. Lines 197-211 was suggested to summarize in a Table. Thank you for the idea that we certainly will follow in a later study when we have more data to present. Here, however, we had few data to present to be worth the labour.

Regards

Round 2

Reviewer 1 Report

All comments addressed by authors

Reviewer 4 Report

Authors' reply was sufficient to my previous comments.  I recommend the revised version for publication.